# Analysis of Differences in Injuries in Padel Players According to Sport-Specific Factors, Level of Physical Activity, Adherence to the Mediterranean Diet, and Psychological Status

**DOI:** 10.3390/sports13070228

**Published:** 2025-07-10

**Authors:** Guillermo Rocamora-López, Adrián Mateo-Orcajada

**Affiliations:** 1Facultad de Fisioterapia, Podología y Terapia Ocupacional, Universidad Católica de Murcia (UCAM), 30107 Murcia, Spain; grocamora@alu.ucam.edu; 2Facultad de Deporte, Universidad Católica de Murcia (UCAM), 30107 Murcia, Spain

**Keywords:** mediterranean diet adherence, padel, physical activity, prevention, psychological state, risk factors, sport characteristics, sport injuries

## Abstract

The available scientific evidence on padel injuries is scarce and inconclusive. For this reason, the main aim was to analyze the differences in injury incidence in padel according to specific factors of the sport, as well as to the level of physical activity, adherence to the Mediterranean diet, and the psychological state of the players. A sample of 216 padel players (mean age: 30.05 ± 9.50 years old) participated in this study. The participants completed a sociodemographic questionnaire that included padel-specific variables, a sports injury questionnaire, the IPAQ, the MEDAS, and the CPRD. A higher incidence of injuries was observed in players with more experience (*p* < 0.001), more hours of play (*p* < 0.001) and at amateur or professional levels (*p* < 0.001). Mild and moderate injuries were common with mixed or herringbone soles; severe (*p* = 0.031), muscle, tendon and ligament injuries were common with herringbone soles (*p* = 0.023). Muscle and ligament injuries occurred more frequently on sand courts (*p* = 0.037), and with 350–370 g racquets (*p* = 0.029). Tendon injuries were associated with less mental ability (*p* = 0.014). There were no significant differences with the Mediterranean diet or level of physical activity. Injury in padel is related to sport-specific factors and psychological state but does not seem to be related to level of physical activity or diet. However, due to the cross-sectional design, causal relationships cannot be established, so future research in this field is needed.

## 1. Introduction

Padel is a racket sport that has experienced significant growth in recent years, becoming highly popular and practiced in various parts of the world [1,2]. One of the main characteristics of this sport is that it is played in pairs on a 10 × 20 m court, which includes side and back walls (Figure 1). The court may be either indoors or outdoors and may contain sand or not on its surface [2]. In addition to the specific court, playing padel only requires a padel racket and official balls, although other equipment such as specific footwear may also be used [2].

The rapid and widespread development of this sport has led to the organization of international competitions around the world, making player performance and the optimization of strokes and movements on the court increasingly important [3]. As a result, scientific research has emerged that analyzes the key elements that influence player performance [4,5], with the aim of identifying and optimizing the most relevant actions and behaviors during play to enhance athletes’ efficiency.

Among the sport-specific factors that appear to influence padel performance, we find: match duration [6], which can vary depending on gender [7], competitive level [8], or the age of the players [7]; the distance covered during a match, which tends to be greater for higher-level players or in more balanced matches [9], and also depends on the number of times a player serves during the match, as servers typically cover more distance than returners [9,10]; the pair’s playing strategy, with the Australian formation requiring the server to cover a greater distance [11]; the playing side, with the backhand-side player being more involved in the final strokes of each point [12,13]; the total number of strokes per point, which increases in points played during the third set of matches [14]; and the hitting position, with players who remain at the back of the court losing a higher percentage of points [12].

In addition to sport-specific factors, differences have been observed in certain physiological, physical, and psychological variables among players in relation to performance. For instance, amateur players spend a large portion of the match engaging in aerobic efforts [15], and it was observed that autonomic modulation was affected after playing matches, with a significant decrease in low frequencies (LF) and an increase in high frequencies (HF) during the match [16]. Significant differences in these values were also found across different time periods following the match (Figure 2) [16]. This is highly relevant, as HF reflects parasympathetic vagal activity associated with heart rate variations linked to the respiratory cycle, while LF reflects both sympathetic and parasympathetic nervous system functions, as well as blood pressure changes, all of which have a considerable impact on athletic performance [17]. In addition, higher-level players tend to spend more time within 50–70% of their maximum heart rate during matches and report a lower perceived exertion afterward [18], as well as a faster recovery capacity following the effort [19]. These higher-level players also perform better in specific physical fitness tests, showing lower heart rates in the Yo-Yo Intermittent Recovery Test (Yo-Yo IR) [20,21,22] and displaying higher levels of lumbar strength compared to amateur players [23].

In the psychological domain, it has been found that playing several consecutive matches increases mental fatigue both before and after competition, while decreasing motivation and reaction time—factors that ultimately reduce performance [24]. Furthermore, players who lost three-set matches exhibited higher levels of anger, fatigue, and depression, and lower levels of vigor compared to those who either won or lost in two sets [25].

In addition to the aforementioned factors, padel is characterized by a high injury incidence, with between 40% and 70% of players sustaining at least one injury per year [26], which significantly impacts player performance. Injury epidemiology varies depending on several factors, such as the body region affected, the moment during the match when the injury occurs, the anatomical structure involved, the athlete’s anthropometric characteristics, playing level, the type of racket used, the previous warm-up or the level of general fitness [27,28,29,30].

Foot, knee, elbow, and shoulder injuries are among the most prevalent in padel [28,30,31], although the affected body region varies depending on the moment within the match. Upper-limb injuries tend to occur more frequently in the final stages of matches, whereas lower-limb injuries are more common at the beginning [29]. Injury severity also differs by region, with upper-limb injuries tending to be mild and lower-limb injuries more often classified as moderate [30]. Additionally, player anthropometric characteristics play a role, with shoulder injuries being more prevalent in taller players and those who play on the backhand side [32,33].

The most commonly affected anatomical structure is the tendon, accounting for 40.4% of all injuries in this sport [27], followed by muscles (30.7%) and ligaments (17.5%) [30]. Regarding severity, muscle and tendon injuries are generally mild, whereas ligament injuries tend to be of moderate severity [30]. In terms of playing level, lower-level players are more prone to tendinopathies and plantar fasciopathies, while higher-level players more frequently experience muscular overload, likely due to the specific stroke mechanics employed [30,34]. The type of racket used also seems to influence injury risk, with players using round-shaped rackets weighing over 350 g being more susceptible to injury [30,35].

Regarding warm-up routines, it has been observed that they are closely related to the likelihood of injuries, particularly in the lower limbs, as these injuries often occur at the beginning of matches and are associated with inadequate warm-up practices [36]. This is because an appropriate warm-up, incorporating balance exercises, dynamic stretching, and sport-specific drills, has been shown to be effective in reducing the injury rate [36], as well as enhancing subsequent athletic performance [37]. Similarly, players’ physical condition appears to be a key factor in injury prevention, as muscular imbalances and poor physical fitness have been associated with an increased risk of injury [19]. This is also evident in recreational players, where the risk of injury is higher among those who do not engage in any type of mobility work, stretching, or sport-specific strength training [38].

All of this indicates that injury incidence in padel is a relevant factor that impacts athlete performance. The scientific literature has attempted to analyze sport-specific factors associated with higher injury rates; however, the existing studies are limited in number and often contain methodological constraints that hinder the generalization of the findings. Moreover, to date, no studies have been identified that examine the influence of player habits on injury incidence in padel.

In this regard, previous research in other sports has shown that factors such as players’ physical activity levels [39,40], adherence to nutritional patterns [41,42], and psychological state [43,44], may be determinants factors in injury occurrence. Specifically, higher levels of physical activity appear to act as a protective factor, being associated with lower injury incidence [39,40]; athletes who adhere to healthy dietary patterns—such as the Mediterranean diet—tend to show better musculoskeletal health [41,42] and may experience improved rehabilitation outcomes [45]. However, a clear relationship between dietary intake and injury occurrence has not been firmly established [46]. Additionally, stress, due to its influence on athletes and other psychological variables [43,44], as well as factors such as concentration, self-confidence, anxiety, and the ability to cope with external evaluations, have all been linked to injury incidence [47,48].

In light of this identified research gap regarding injury incidence in padel, the present study aims to analyze differences in injury incidence (including occurrence, severity, progression, and anatomical structure), based not only on sport-specific factors, but also on the players’ physical activity levels, adherence to the Mediterranean diet, and psychological state.

## 2. Materials and Methods

### 2.1. Design

A study using a descriptive observational design was conducted using non-probabilistic convenience sampling. This study followed the STROBE guidelines [49] for the design and reporting of observational research. Prior to commencement, this study was approved by the Institutional Ethics Committee of the Universidad Católica San Antonio de Murcia (UCAM) (code: CE012519) and was conducted in accordance with the principles of the Declaration of Helsinki. All participants were informed of the study procedures and provided their written informed consent before participating in the research.

### 2.2. Participants

A total of 223 participants from were initially included in the study sample. After applying the inclusion and exclusion criteria, the final sample consisted of 216 participants (mean age: 30.05 ± 9.50 years), comprising 163 men and 53 women. Figure 3 shows the sample flow diagram.

On average, participants played three hours of matches per week (SD: 2.29), engaged in one and a half hours of weekly training (SD: 2.55), and had five and a half years of experience in the sport (SD: 6.39). Table 1 shows the sample’s playing characteristics, and Table 2 summarizes injury-related characteristics.

The inclusion criteria were: (a) participants aged between 18 and 65 years old; and (b) individuals who played padel regularly. The exclusion criteria were: (a) individuals under 18 or over 65 years of age (*n* = 5); and (b) individuals who did not fully complete the questionnaires (*n* = 2).

The sample size calculation was performed using the RStudio statistical software (version 3.15.0; Posit PBC, Boston, MA, USA). Standard deviations (SD) from previous studies on injury incidence in padel (SD = 0.55) were used as a reference [50]. Based on a margin of error (*d*) of 0.08 for weekly match hours, and a 95% confidence interval (CI), the minimum required sample size for this study was 173 players.

### 2.3. Instruments

To collect the sociodemographic variables, as well as variables related to the practice of padel, an ad hoc questionnaire was used, which was developed following the guidelines of Carretero-Dios et al. [51].

Perceived physical activity level was assessed using the International Physical Activity Questionnaire (IPAQ). This instrument consists of seven questions regarding the frequency, duration, and intensity of physical activity performed during the seven days prior to completing the questionnaire [52]. The unit of measurement used is the Metabolic Equivalent of Task (MET), which allows for the classification of physical activity levels as low, moderate, or high. Individuals classified as moderate or high are considered to meet the World Health Organization (WHO) physical activity recommendations. This questionnaire has been validated in previous studies and has shown a Cronbach’s alpha of 0.928 [52].

Sport-related injuries were recorded using the questionnaire developed by García-Fernández et al. [30], which collects information on injury incidence, affected body region, type of tissue involved, nature of the injury, and the causes and severity of the injury.

Adherence to the Mediterranean diet was assessed using the Mediterranean Diet Adherence Screener (MEDAS), which consists of 14 questions related to dietary habits [53]. The responses are recorded based on agreement or disagreement with each statement, with scores of 1 or 0, respectively. The total score ranges from 0 to 14, with higher scores indicating greater adherence to the Mediterranean diet. Scores below 9 are considered low adherence, while scores above 9 indicate good adherence. This questionnaire has been used and validated in previous studies and shows an internal consistency coefficient of 0.51 [53].

The Psychological Characteristics Related to Sport Performance questionnaire (CPRD) was used to assess the influence of psychological variables on the athletic performance of the participants. It demonstrates an internal consistency coefficient of 0.85 [54]. The questionnaire consists of 55 items grouped into five factors: stress control, with a maximum score of 80 (Cronbach’s alpha = 0.88); influence of performance evaluation, maximum score of 45 (Cronbach’s alpha = 0.72); motivation, maximum score of 31 (Cronbach’s alpha = 0.67); mental skills, maximum score of 34 (Cronbach’s alpha = 0.34); and team cohesion, maximum score of 24 (Cronbach’s alpha = 0.78). Higher scores on each factor indicate that the participant possesses better psychological tools or skills to cope with situations. The responses were recorded on a Likert scale from 1 (strongly disagree) to 5 (strongly agree). Subsequently, the scores for the items within each category were reverse scored for the final calculation, following the recommendations from previous research [54].

### 2.4. Procedure

First, padel clubs in the Region of Murcia and the Valencian Community (Spain) were contacted to inform them about this study and its procedures. After obtaining their approval, an informational message regarding this study was distributed to the members of these clubs who agreed to participate. Interested players completed the informed consent form and subsequently received the questionnaire digitally for completion.

In addition to assisting with player recruitment, the padel clubs in the Region of Murcia and the Valencian Community facilitated contact between participants and the researchers in cases of questions regarding questionnaire completion.

### 2.5. Data Analysis

The normality of the variables was assessed using the Kolmogorov–Smirnov test, as well as skewness and kurtosis analyses. Since the variables followed a normal distribution, parametric tests were conducted for analysis. Means and standard deviations (M ± SD) were used as descriptive statistics, and percentages were employed for frequency analysis. An independent-samples Student’s t-test was performed to analyze differences in the study variables between players who had suffered an injury and those who had not. Cohen’s d was used to determine effect size [55], and was interpreted as small (*d* = 0.10–0.29), moderate (*d* = 0.30–0.49), large (*d* = 0.50–0.69), or very large (*d* > 0.70). Three one-way ANOVAs were performed to analyze differences in the study variables according to injury severity, injury evolution, and type of injured structure. Partial eta squared (η^2^) was used to calculate effect size, defined as small (ES ≥ 0.10), moderate (ES ≥ 0.30), large (ES ≥ 1.2), or very large (ES ≥ 2.0) [56]. Bonferroni post hoc tests were applied to identify significant differences between groups. Four chi-square analyses were performed to examine differences in categorical variables. Cramer’s V was used as a post hoc comparison to assess the strength of association between variables, ranging from 0 to 1. A *p*-value of <0.05 was set to determine statistical significance. The statistical analyses were performed using SPSS software version 26 (SPSS Inc., Chicago, IL, USA).

## 3. Results

### 3.1. Analysis of Differences According to Injury Occurrence

Table 3 and Figure 4 present the differences in study variables according to injury occurrence. Significant differences were found in relation to weekly hours of matches, with players who spent more time playing matches experiencing a higher incidence of injury (*p* = 0.001). Differences were also observed in years of experience, with injured players having more years of experience (*p* < 0.001). It is worth noting that the effect size was very large in both cases (match hours: *d* = 2.24; years of experience: *d* = 6.18). No significant differences were found in injury occurrence with respect to weekly hours of training (*p* = 0.780), physical activity level (*p* = 0.880), adherence to the Mediterranean diet (*p* = 0.060), or psychological status of the players (*p* = 0.110–0.960) (Table 3).

Additionally, differences were also found based on the level of play, with more injuries occurring among amateur and professional or semi-professional players, with a large effect size (*p* < 0.001; X^2^ = 23.73; effect size: 0.331). No significant differences were observed in injury occurrence related to court surface (*p* = 0.333; X^2^ = 0.94; effect size: 0.066), racket weight (*p* = 0.568; X^2^ = 3.87; effect size: 0.162) or footwear used (*p* = 0.084; X^2^ = 6.66; effect size: 0.176) (Figure 4).

### 3.2. Analysis of Differences According to Injury Severity

Table 4 and Figure 5 show the differences according to injury severity. No significant differences were found based on weekly hours of matches or training (*p* = 0.307–0.456), years of experience (*p* = 0.106), physical activity level (*p* = 0.235), adherence to the Mediterranean diet (*p* = 0.979), or psychological variables (*p* = 0.163–0.999) (Table 4).

Significant differences were found regarding the type of footwear used, with mild and moderate injuries occurring mainly in players who used shoes with mixed or herringbone soles, and severe injuries in those who used herringbone soles, although with a moderate effect size (*p* = 0.031; X^2^ = 13.84; effect size: 0.245). No significant differences were observed in relation to court type (*p* = 0.801; X^2^ = 0.44; effect size: 0.062), racket weight (*p* = 0.520; X^2^ = 9.13; effect size: 0.224), or level of play (*p* = 0.768; X^2^ = 1.82; effect size: 0.768) (Figure 5).

### 3.3. Analysis of Differences According to Injury Progression

Differences in the study variables according to injury changes are presented in Table 5 and Figure 6. Significant differences were found in players’ years of experience (*p* = 0.038), specifically between acute and chronic injuries (*p* = 0.040), with chronic injuries occurring more frequently in players with more years of experience. However, the effect size found for these differences was small (effect size: 0.06). No significant differences were found in the other variables based on injury type (*p* = 0.306–0.963) (Table 5).

No significant differences were found in the injury progression regarding court surface (*p* = 0.228; X^2^ = 2.96; effect size: 0.158), racket weight (*p* = 0.516; X^2^ = 9.17; effect size: 0.223), footwear type (*p* = 0.454; X^2^ = 5.74; effect size: 0.157), or playing level (*p* = 0.142; X^2^ = 6.89; effect size: 0.172) (Figure 6).

### 3.4. Analysis of Differences According to the Injured Anatomical Structure

Table 6 and Figure 7 present the differences in study variables according to the injured anatomical structure. Significant differences were found in years of experience, with more bone injuries occurring in players with greater experience, while ligament and muscle injuries were more common in less experienced players (*p* = 0.049). Similarly, differences were observed in players’ mental skills (*p* = 0.020). Specifically, players with tendon injuries showed lower mental skills as compared to those with muscle injuries (*p* = 0.014). The effect size was small in both cases (years of experience: 0.10; mental ability: *p* = 0.11). No significant differences were found based on weekly match hours (*p* = 0.659), training hours (*p* = 0.258), physical activity level (*p* = 0.985), or adherence to the Mediterranean diet (*p* = 0.875) (Table 6).

The type of court showed differences in the structure affected by the injury. Muscle and ligament injuries occurred more frequently on sand courts, while tendon and bone injuries occurred on both sand and non-sand courts (*p* = 0.037; X^2^ = 32.06; effect size: 0.224). Regarding racket weight, most muscle, tendon and ligament injuries occurred in players using rackets weighing between 350 and 370 g (*p* = 0.029; X^2^ = 39.99; effect size: 0.296). The footwear used also showed significant differences, with most muscle, tendon and ligament injuries occurring in players wearing mixed or herringbone sole shoes (*p* = 0.023; X^2^ = 27.77; effect size: 0.284). Concerning the level of play, a greater number of muscle, tendon, and ligament injuries occurred in amateur and semi-professional or professional players (*p* = 0.004; X^2^ = 25.68; effect size: 0.334). The effect size was moderate for court type, racket weight, and footwear used, but large for playing level (Figure 7).

## 4. Discussion

The objective of the present study was to analyze the differences in injury incidence (occurrence, severity, progression, and anatomical structure) in players, considering factors inherent to the sport, as well as the players’ physical activity level, adherence to the Mediterranean diet, and psychological status.

### 4.1. Differences in the Study Variables According to Injury Occurrence

Significant differences were found in injury incidence based on the number of weekly hours dedicated to matches, but not for training hours, with players who spent more time playing matches showing a higher occurrence of injuries. This result showed a very large effect size, indicating a highly significant difference that is also consistent with previous research that observed a tendency for injuries to increase with more match hours, although the differences were not statistically significant [30,31,57]. A possible explanation for these findings is that the demands of competitive padel, similar to other sports, are greater than those of training sessions, significantly increasing the risk of injury due to continuous accelerations, decelerations, and changes in direction [58]. In this regard, implementing training sessions under pressures that simulate competitive demands could help players better prepare by more accurately replicating match conditions [59].

A higher injury incidence was also observed in players with more years of experience, with a very large effect size. In this case, previous scientific literature reports contradictory findings, with some authors indicating that injury incidence increases with greater player experience [30,31], while others report a higher prevalence of injuries among players with less than five years of experience [35]. An increased number of years of experience is typically associated with a greater volume of training and match hours, as well as a higher number of strokes performed, which may explain the findings observed in the present study. However, years of experience alone do not appear to be the most decisive factor for injury occurrence, as evidenced by the divergent results in previous investigations [30,31,35]. Therefore, it would be pertinent for future research to analyze years of experience in conjunction with other factors related to injury incidence, such as physical condition, playing intensity, and hours of training and competition, to more accurately determine the relevance of experience in injury occurrence in padel.

Regarding the players’ level of play, a higher incidence of injuries was observed among amateur and professional, or semi-professional players as compared to beginners, with a large effect size for this findings. There is no consensus in the scientific literature on injury incidence and playing level in padel [30], as some studies report a higher injury rate among lower-level players [26], whereas others present findings similar to the current study, indicating that injuries are more frequent at higher competitive levels [60]. In fact, even among high-level competitive players, those ranked higher in the classification tend to experience more sports injuries [60]. A possible explanation for the lack of agreement between competitive level and injury incidence is that lower-level players tend to compensate for their lack of technique with greater physical effort, while higher-level players perform a larger proportion of overhead and high-velocity strokes, increasing the risk of upper-limb injuries [13,61]. Therefore, competitive level appears to be relevant in injury occurrence, but this is also associated with other padel-specific factors that should be considered in future research.

Regarding physical activity level, adherence to the Mediterranean diet, and the psychological status of padel players, no significant differences were found in relation to injury occurrence. This contrasts with previous studies conducted in other sports disciplines, where a low level of physical activity was associated with a higher injury incidence [39,40]; similarly, poorer adherence to the Mediterranean diet was linked to increased injury risk [42]; and athletes with poorer psychological status, possessing fewer coping tools for stressful situations, suffered more injuries [44,47].

### 4.2. Differences in the Study Variables According to Injury Severity

In the present study, mild and moderate injuries predominated, which is consistent with previous research reporting that most injuries were mild [57]. Regarding the factors that showed significant differences in injury severity, the type of footwear used stands out, with mild and moderate injuries occurring mainly in those who wore shoes with mixed or herringbone soles, and severe injuries in those who wore herringbone soles, with a moderate effect size for this finding. Previous studies have shown that recreational players who use padel-specific footwear have a higher injury rate than those who do not use this type of footwear [62]. The sports equipment used in padel, as in other sports, is fundamental due to its influence on injury incidence. This has also been observed with other equipment such as the racket, since its characteristics or improper use have been shown to be determinants in injury occurrence, given that not all players have the same characteristics [35,63]. Regarding footwear, no studies have analyzed the impact of shoe type on injury severity in padel, but in tennis, the lower torsional stiffness of footwear has been shown to increase forefoot inversion angle and may reduce the risk of ankle sprain during forehand strokes, whereas higher stiffness may increase risk during open defensive forehand positions [64]. Additionally, it has been observed that the type of footwear significantly influences lower-limb kinematics and internal loading conditions during rapid lateral movements, which could generate pain and increase injury risk [65]. Therefore, although these are different sports, the footwear-related risk factors shown to be important in tennis could also be relevant in padel. Selecting appropriate footwear, even if padel-specific, is crucial for reducing injury probability as well as injury severity. The results obtained are relevant, given that although previous studies have shown a relationship between sole type and injury incidence [30], no studies have analyzed the relationship between sole type and injury severity.

No significant differences were found in injury severity based on physical activity level, adherence to the Mediterranean diet, or the psychological status of padel players. This contrasts with previous research conducted in other sports disciplines, where low physical activity levels were associated with greater injury severity [66]; or where a better management of performance evaluation was related to fewer severe injuries as compared to athletes with poorer management of this evaluation [47]. However, the results are similar to those from other studies regarding adherence to the Mediterranean diet, as it was also not related to injury severity [46].

### 4.3. Differences in the Study Variables According to Injury Progression

The results showed significant differences according to the players’ years of experience, with chronic injuries occurring more frequently in players with more experience. These findings are consistent with previous studies that reported acute and chronic injuries as the most common in padel [57], while recurrences, although occurring occasionally, were less frequent [57]. It is worth noting that no studies have analyzed differences in changes in injury based on years of experience within the context of padel. A possible explanation for these results is that the most common injuries in padel are tendinopathies, specifically epicondylitis, which is a long-lasting condition where greater exposure time to the repetitive stimulus causing the injury increases the risk of developing a chronic injury [27,34]. However, the effect size for this finding was small, and the sample size within each group was also limited. Therefore, these results should be interpreted with caution, and further research is needed before drawing general conclusions.

No significant differences were observed in injury changes according to the level of physical activity, adherence to the Mediterranean diet, or the psychological status of padel players. These results differ from previous studies conducted in other sports disciplines, where low physical activity levels were associated with a higher injury incidence [66]. Regarding the Mediterranean diet, although there are no studies with athletes linking it directly to injury changes, it is well established that this dietary pattern plays an important role in bone health [67], potentially benefiting athletes at risk of chronic injuries such as osteoporosis or stress fractures [68,69]. Furthermore, previous research has shown that injured athletes with lower psychological coping abilities tend to experience more injuries as compared to those with better coping skills, thereby increasing the risk of recurrence [47].

### 4.4. Differences in the Study Variables According to Injured Anatomical Structure

Significant differences were found according to years of experience, with more experienced players presenting a higher incidence of bone injuries, while less experienced players exhibited more ligament and muscle injuries. Most bone injuries are stress fractures, primarily caused by prolonged exposure to microtraumas, which logically suggests that more experienced players, having a greater exposure to these microtraumas, sustain more bone injuries [70]. Additionally, although this association does not apply to all cases, less experienced players tend to be younger, and previous research has shown that younger players suffer more ligament injuries [26], which is consistent with the findings of the present study. Another possible explanation could be related to the players’ technique, as differences in the execution of sports movements, such as the smash, have been observed according to experience level [71]. Due to the small effect size observed for this finding, future research should analyze whether years of experience are truly a determining factor in the type of injury, including larger and more diverse samples to determine its consistency and potential, as well as whether other aspects more related to technique or time of exposure to the sport are more influential.

A relevant finding was the differences observed according to the players’ mental skills. Specifically, players with tendon injuries exhibited lower mental skills than those with muscle injuries. Little is known about the influence of psychological variables on padel injuries beyond the indication that stress appears to be an influential factor [50]. However, studies on tennis suggest that players with fewer muscle injuries demonstrate a greater control over sport-related stress; those with better management of performance evaluations suffer fewer tendinopathies; and those with fewer fractures exhibit higher levels of team cohesion [72]. This highlights the important role played by different psychological variables on performance and injury risk in racket sports. Furthermore, mental fatigue has been shown to affect the psychomotor performance specific to padel players, particularly impacting all types of stroke execution and reaction time [73]. Just as mental fatigue influences technique, poorer mental skills could similarly do so, leading to improper strokes and an increased likelihood of sustaining tendon injuries such as epicondylitis or rotator cuff involvement, which are the most common in racket sports [30]. On the other hand, poorer mental skills could be associated with suboptimal training planning, difficulties in load self-regulation, or a lower adherence to recovery strategies, factors that contribute to the progressive development of overuse injuries such as tendinopathies, whereas muscle injuries tend to be more directly related to direct trauma or tears [74]. Despite these findings, it should be taken into consideration that this was the only psychological variable that showed significant results but had a small effect size. Considering that the questionnaire used shows a reduced reliability for this dimension, it would be essential that future research addresses the psychological aspect with other techniques such as the interview in order to obtain a real vision of the relationship between injuries and psychological state in padel tennis players.

It was also observed that muscle and ligament injuries occur more frequently in sand courts, whereas tendon and bone injuries occur in both sand and non-sand courts. Only one padel study has associated court type with injury probability, indicating that players are more prone to maxillofacial injuries due to the racket rebounding off the glass wall, causing skin wounds, injuries, or fractures [75]. However, in other racket sports such as tennis, it has been observed that the type of court can affect injury incidence, as the characteristics of play differ depending on the surface [76]. In padel, these results could be related to the biomechanical adaptations of the player to the court type. Sand surfaces, by offering greater braking capacity and traction, generate higher eccentric demands during explosive movements and sudden stops, which could increase the risk of muscle overload injuries or eccentric contractions from large stretching [77]. Conversely, non-sand courts, being faster and more slippery, may promote increased sustained mechanical tension on tendons, and lower impact absorption capacity, which would explain the presence of tendon injuries on both types of surfaces [77]. This information highlights the importance of considering the playing surface when planning training and competition periods to prevent injuries. This is because at present, professional players mainly compete on non-sand courts, while amateur players regularly switch playing surfaces, conditioning injury occurrence during practice. Given that the effect size for this finding was moderate, the relevance of this result should be considered, considering that the continuous change in playing surfaces may lead to a higher risk of injury due to the different physical demands each surface places on the body’s structures.

Regarding sports equipment, the factors of racket and footwear used showed significant differences in the affected anatomical structure. With respect to racket weight, most muscle and tendon injuries occur in players using rackets weighing between 350 and 370 g. Previous studies have shown an increased injury incidence in amateur players using rackets heavier than 350 g [35]. The results provided by the present study go further and demonstrate the relationship between racket weight and the affected anatomical structure. The main reason for these differences could be that with heavier rackets, a greater effort is exerted in technical gestures, thus affecting the entire elbow and shoulder complex, potentially causing greater muscle fatigue and tendinopathies of both the epicondylar extensors and the rotator cuff due to overuse [34]. The relevance of this finding lies in the fact that, during a match, approximately 1000 ball strikes are performed by each player [78]. If the racket weight is inadequate and not properly supported by the body’s structures, this repetitive load may lead to progressive wear, ultimately resulting in injury.

According to the footwear used, most muscle, tendon and ligament injuries occurred in players wearing mixed and herringbone sole shoes, showing this finding a moderate effect size. Previous studies [30] observed that OMNI-type shoes predisposed the players to a higher incidence of lower-limb injuries, although no relationship was found regarding the injured anatomical structure. The differences found in the present study could be explained by the mechanical behavior of each sole type during play. Herringbone sole footwear, besides being the most widely available on the market, is commonly recommended for sanded surfaces, as it provides a high traction that can favor explosive movements, but may also generate greater tension on muscle and ligamentous structures if the foot remains fixed during sudden changes in direction [77]. On the other hand, mixed sole footwear, by attempting to adapt to different surface types, may cause instability or a lack of grip in certain game situations, generating excessive tension and load to compensate for this lack of grip, thus increasing the risk of tendinopathies or muscle injuries [77]. Therefore, the choice of specific padel footwear may condition the injured anatomical structure, although further research is needed to clarify this information and provide stronger scientific evidence.

Regarding the level of play, a higher number of muscle, tendon, and ligament injuries occur in amateur and semi-professional or professional players as compared to beginners, showing a large effect size. Sánchez-Alcaraz et al. [27] observed that amateur players have a higher incidence of tendon and ligament injuries, which could be attributed to a lower physical fitness or poorer sports technique in strokes such as the backhand, increasing the risk of epicondylitis [79]. Additionally, previous studies have shown that player positioning at the moment of striking the ball differs between amateur players with some training experience and beginners. Specifically, trained players hit the ball from a more delayed position during serve and offensive strokes, but use more advanced and closer-to-the-ball positions during defensive shots [80]. This has been demonstrated to be important when designing training plans and injury prevention strategies. It has also been observed that the types of strokes used vary according to the level of the player; amateur players are characterized by using forehands, backhands, wall rebounds, and powerful smashes, while professionals employ shots to the fence, backhand volleys, drop shots, “víboras”, or smashes [81]. This distinction is crucial because it influences the musculoskeletal demands during training and competition. This information is highly relevant, as it highlights the importance of a proper injury prevention program that is individualized for each player, focusing on load management in more experienced players and improving technique and physical preparation in less experienced ones.

No significant differences were observed in the injured structure according to the level of physical activity, which aligns with the findings by Castillo-Lozano and Casuso-Holgado [28], who also found no significant differences when analyzing a sample of padel players. Likewise, no significant differences were found regarding adherence to the Mediterranean diet, which contrasts with other studies that highlighted the importance of this nutritional pattern in the prevention and rehabilitation of bone injuries such as fractures [69]. Despite the results found regarding the affected anatomical structure, and the fact that the effect sizes range from moderate to large, it is important to consider that the established subdivisions resulted in small sample sizes for some comparisons. This may have influenced the observed results and highlights the need for future studies to confirm these findings.

### 4.5. Limitations

This study is not without limitations. This study used a cross-sectional design, which limits the ability to infer causality. Therefore, while associations between variables can be identified, it is not possible to determine directional or causal relationships. Future longitudinal or experimental studies are needed to confirm these findings and explore causal mechanisms. Second, the number of males was much higher than that of females, so an analysis based on gender could not be conducted. Third, some subgroup analyses, such as those involving bone injuries, were based on a very small number of cases and therefore lacked sufficient statistical power. As a result, these findings should be interpreted with caution and cannot be generalized. Fourth, although the questionnaires used are valid and reliable, there remains a risk of bias derived from the participants’ subjective perception and a possible misinterpretation of the questions. Regarding the questionnaires, some of the tools used in this study presented low internal consistency. This may limit the reliability of the data obtained from these instruments. Future studies should consider the use of additional or alternative tools to obtain more information and complement the results obtained in the present investigation. Fifth, being a retrospective study, it is limited by the fact that the information collected depends on previous records and participants’ memory, which may have affected data accuracy. Sixth, the lack of control over some external variables, such as players’ training levels or types of previous injuries, may have also influenced the interpretation of the results. And seventh, the sample was obtained through convenience sampling from federated padel players in Murcia and Valencia (Spain), which may introduce a regional or cultural bias. Therefore, caution should be exercised when generalizing the results to other populations or contexts.

## 5. Conclusions

Players who dedicate more hours to matches, as well as those with more years of experience, present a higher number of injuries as compared to those with fewer match hours and less experience. Concerning the level of the players, more injuries are observed in amateur and professional, or semi-professional players as compared to beginners.

Regarding injury severity, more mild and moderate injuries are observed in players using shoes with a herringbone sole, but no significant differences are found concerning other game variables or when considering physical activity level, adherence to the Mediterranean diet, or the players’ psychological status. In terms of injury progression, chronic injuries are more frequent in players with more years of experience.

Regarding the affected anatomical structure, more bone injuries occur in players with more years of experience, while those with fewer years of experience have more ligament and muscle injuries. Amateur, semi-professional, and professional players present more muscle, tendon, and ligament injuries than beginners. Concerning padel-specific factors, more muscle and ligament injuries occur in courts with sand, whereas tendon and bone injuries occur in both courts with and without sand; racket weight is relevant, with muscle, tendon and ligament injuries being more frequent in players using rackets weighing between 350 and 370 g; similarly, footwear influences injury type, with more muscle, tendon and ligament injuries associated with mixed and herringbone sole shoes.

Regarding external factors unrelated to the game, psychological status showed significant differences when considering the affected anatomical structure: specifically, players with tendon injuries had lower mental skills than those with muscle injuries. However, no significant differences were found in psychological variables regarding injury occurrence, severity, or progression, nor in physical activity levels or adherence to the Mediterranean diet, concerning injury in padel players.

Therefore, the results of the present study suggest that injury incidence, severity, progression, and the affected anatomical structure, are mainly related to factors intrinsic to the game of padel, but less so to external factors such as physical activity level or adherence to the Mediterranean diet, although the players’ psychological status does seem to be somewhat relevant. These findings should be interpreted with caution due to the cross-sectional nature of this study, which prevents the establishment of causal relationships. To strengthen these results and explore causality, future research should consider prospective study designs involving larger and more diverse samples.

## Figures and Tables

**Figure 1 sports-13-00228-f001:**
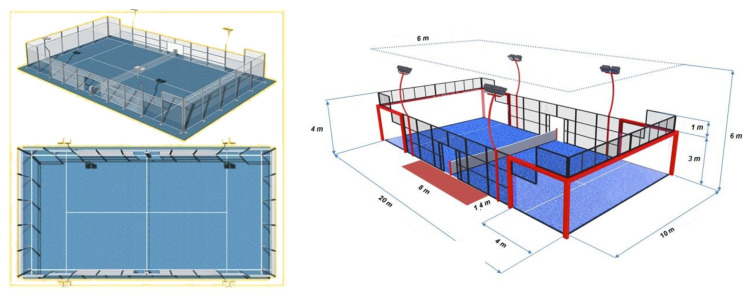
Official padel court.

**Figure 2 sports-13-00228-f002:**
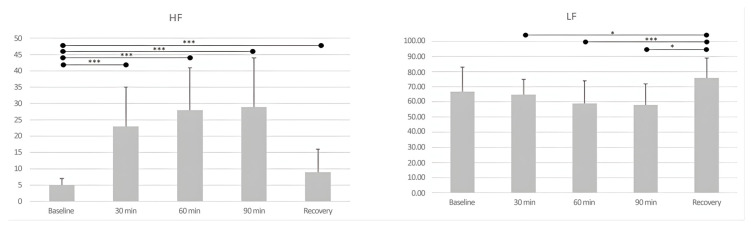
Differences in LF and HF during padel matches and at various time points after the match [16]. *p* < 0.05 *, *p* < 0.001 ***.

**Figure 3 sports-13-00228-f003:**
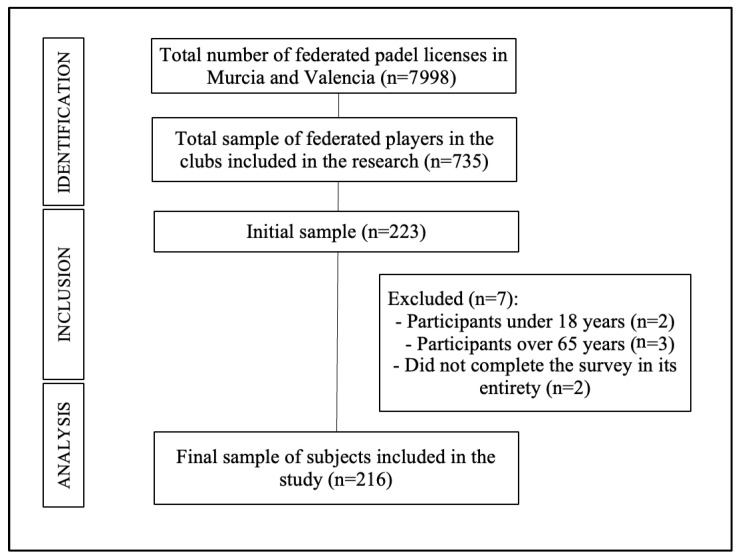
Sample flow diagram.

**Figure 4 sports-13-00228-f004:**
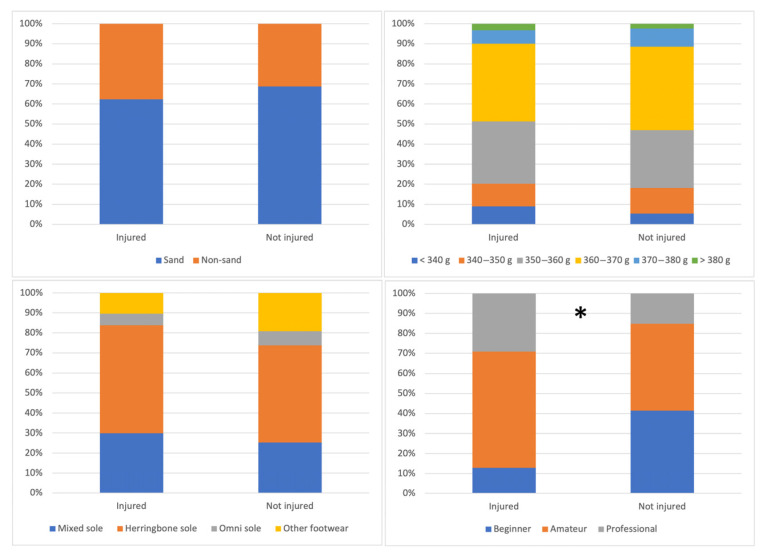
Differences in the occurrence of injury depending on the court surface, racket weight, footwear type and level of play of the padel players. * There are significant differences in this variable between the study groups.

**Figure 5 sports-13-00228-f005:**
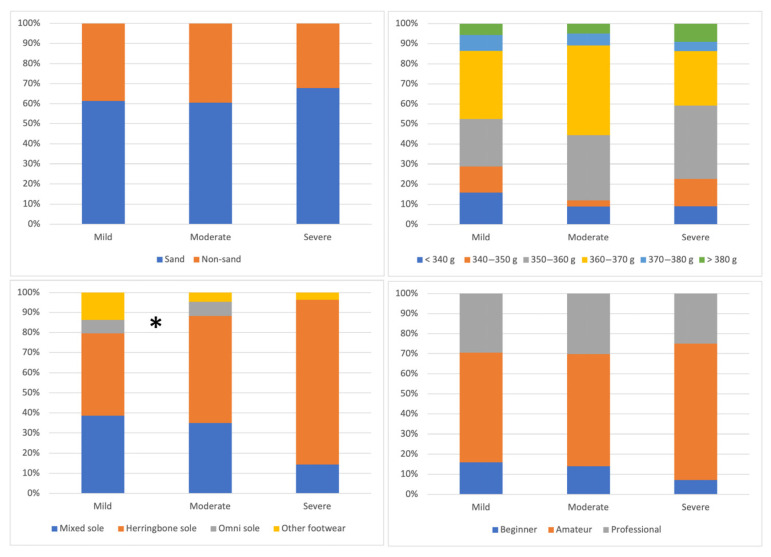
Differences in the severity of the injury depending on the type of court, weight of the racket, shoes used, and level of play of the padel players. * There are significant differences in this variable between the study groups.

**Figure 6 sports-13-00228-f006:**
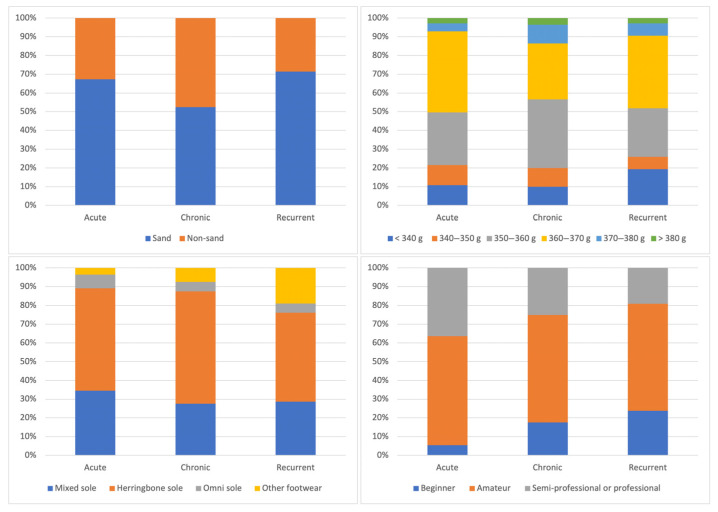
Differences in the injury progression depending on the type of court, weight of the racket, shoes used, and level of play of the padel players.

**Figure 7 sports-13-00228-f007:**
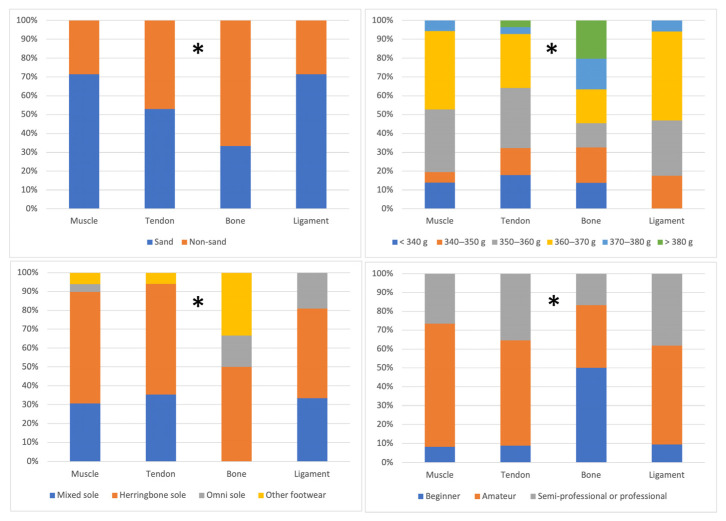
Differences in the type of injured structure depending on the type of court, weight of the padel racket, shoes used, and level of play of the padel players. * There are significant differences in this variable between the study groups.

**Table 1 sports-13-00228-t001:** Characteristics of padel tennis players.

Variable		*n* (%)
Dominant hand		
	Right	199 (92.10%)
	Left	17 (7.90%)
Court position		
	Forehand side	103 (47.70%)
	Backhand side	113 (52.30%)
Court surface		
	Sand	141 (65.28%)
	Non-sand	75 (34.72%)
Racket weight (g)		
	<340 g	14 (9.50%)
	340–350 g	17 (11.60%)
	350–360 g	44 (29.90%)
	360–370 g	58 (39.50%)
	370–380 g	11 (7.50%)
	>380 g	3 (2.00%)
Footwear type		
	Mixed sole	61 (28.20%)
	Herringbone sole	113 (52.30%)
	Omni sole	14 (6.50%)
	Other footwear	28 (13.00%)
Playing level		
	Beginner	56 (25.90%)
	Amateur	111 (51.40%)
	Semi-professional or professional	49 (22.70%)

**Table 2 sports-13-00228-t002:** Injury characteristics of padel players.

Variable		*n* (%)
Injury while playing padel		
	Yes	117 (54.20%)
	No	99 (45.80%)
Injured anatomical structure		
	Muscle	49 (43.60%)
	Tendon	34 (30.60%)
	Bone	6 (6.50%)
	Ligament	21 (19.30%)
Injury severity		
	Mild	44 (38.30%)
	Moderate	43 (37.40%)
	Severe	28 (24.30%)
Injury location		
	Knee	20 (18.20%)
	Elbow	11 (10.50%)
	Shoulder	16 (14.80%)
	Arm	6 (6.20%)
	Ankle/Foot	18 (16.50%)
	Leg and thigh	19 (16.60%)
	Wrist	8 (6.90%)
	Forearm	5 (4.30%)
	Lower back	4 (3.40%)
	Face	3 (2.60%)
Injury progression		
	Acute	55 (47.40%)
	Chronic	40 (34.50%)
	Recurrent	21 (18.10%)

**Table 3 sports-13-00228-t003:** Analysis of differences in the occurrence of injury as a function of time spent in matches and training, years of sporting experience, level of physical activity, level of adherence to the Mediterranean diet, and psychological profile of the players.

Variable	Injured	Not Injured	Mean Diff.	t	*p*	95% CI	*d*
Match hours	3.91 ± 2.50	2.86 ± 1.89	1.05	3.43	0.001	0.44; 1.65	2.24
Training hours	1.51 ± 1.88	1.41 ± 3.17	0.10	0.28	0.780	−0.59; 0.79	2.55
Years of experience	7.12 ± 7.62	3.74 ± 3.82	3.38	4.01	<0.001	1.72; 5.05	6.18
Physical activity (METS)	7687.78 ± 2327.76	7314.07 ± 4699.16	373.71	0.15	0.880	−4502.99; 5250.41	1.51
AMD	8.10 ± 1.64	8.53 ± 1.68	−0.42	−1.87	0.060	−0.87; 0.02	1.66
Psychological variables							
Stress control	49.37 ± 11.43	49.89 ± 12.54	−0.52	−0.32	0.751	−3.74; 2.70	11.95
Influence of performance evaluation	30.49 ± 8.31	32.30 ± 8.00	−1.82	−1.63	0.114	−4.02; 0.38	8.17
Motivation	17.22 ± 5.27	17.26 ± 5.55	−0.04	−0.06	0.963	−1.49; 1.41	5.40
Mental ability	19.14 ± 5.29	19.45 ± 5.27	−0.32	−0.44	0.666	−1.74; 1.10	5.28
Team cohesion	14.95 ± 5.90	14.85 ± 5.92	0.10	0.12	0.907	−1.49; 1.69	5.91

**Table 4 sports-13-00228-t004:** Analysis of differences in injury severity as a function of time spent in matches and training, years of sports experience, level of physical activity, level of adherence to the Mediterranean diet and psychological profile of the players.

Variable	Mild	Moderate	Severe	F	*p*	Effect Size
Match hours	3.61 ± 2.03	3.85 ± 2.13	4.38 ± 3.57	0.790	0.456	0.01
Training hours	1.31 ± 1.43	1.36 ± 1.42	1.96 ± 2.89	1.193	0.307	0.02
Years of experience	5.59 ± 5.20	7.02 ± 4.86	9.50 ± 12.55	2.287	0.106	0.04
Physical activity (METS)	12,352.91 ± 3732.92	4413.65 ± 2854.74	5126.89 ± 5859.34	1.466	0.235	0.03
AMD	8.14 ± 1.71	8.19 ± 1.53	8.11 ± 1.66	0.022	0.979	0.00
Psychological variables						
Stress control	51.36 ± 12.26	48.53 ± 11.52	47.18 ± 10.06	1.284	0.281	0.02
Influence of performance evaluation	32.02 ± 8.89	30.09 ± 7.24	28.46 ± 8.57	1.659	0.195	0.03
Motivation	17.16 ± 5.75	16.53 ± 5.10	18.46 ± 4.77	1.140	0.323	0.02
Mental ability	18.36 ± 5.27	19.07 ± 5.07	20.79 ± 5.51	1.844	0.163	0.03
Team cohesion	14.95 ± 6.51	15.00 ± 6.20	14.93 ± 4.52	0.001	0.999	0.00

**Table 5 sports-13-00228-t005:** Analysis of the differences in the changes in injury as a function of time spent in matches and training, years of sports experience, level of physical activity, level of adherence to the Mediterranean diet, and psychological profile of the players.

Variable	Acute	Chronic	Recurrent	F	*p*	Effect Size
Match hours	3.99 ± 2.07	3.98 ± 3.19	3.50 ± 2.12	0.318	0.728	0.01
Training hours	1.37 ± 1.38	1.60 ± 2.57	1.64 ± 1.54	0.238	0.788	0.00
Years of experience	5.71 ± 5.08 *	9.63 ± 10.87 *	6.19 ± 4.18	3.366	0.038 *	0.06
Physical activity (METS)	9989.20 ± 3258.75	4433.23 ± 5071.63	7529.76 ± 1363.14	0.651	0.523	0.01
AMD	8.07 ± 1.80	8.15 ± 1.27	8.19 ± 1.86	0.048	0.953	0.00
Psychological variables						
Stress control	51.05 ± 11.52	47.58 ± 11.61	48.19 ± 10.94	1.197	0.306	0.02
Influence of performance evaluation	30.75 ± 7.95	30.05 ± 9.74	30.95 ± 6.50	0.110	0.896	0.00
Motivation	17.35 ± 4.95	17.15 ± 5.50	16.90 ± 5.97	0.054	0.947	0.00
Mental ability	19.07 ± 4.72	18.98 ± 5.46	19.76 ± 6.56	0.165	0.848	0.00
Team cohesion	14.96 ± 5.36	14.73 ± 6.27	15.14 ± 6.84	0.037	0.963	0.00

* Differences are significant at the *p* < 0.05 level.

**Table 6 sports-13-00228-t006:** Analysis of differences in the type of injured structure as a function of time spent in matches and training, years of sporting experience, level of physical activity, level of adherence to the Mediterranean diet, and psychological profile of the players.

Variable	Muscle	Tendon	Bone	Ligament	F	*p*	Effect Size
Match hours	3.62 ± 1.74	4.00 ± 3.27	3.17 ± 2.56	4.67 ± 2.76	0.654	0.659	0.03
Training hours	1.47 ± 1.50	1.20 ± 1.63	3.25 ± 5.06	1.45 ± 1.49	1.328	0.258	0.06
Years of experience	6.47 ± 4.89 *	8.15 ± 6.17	15.83 ± 25.28 *	5.57 ± 4.72 *	2.305	0.049	0.10
Physical activity (METS)	9311.10 ± 3438.34	5921.38 ± 6126.37	4251.50 ± 2166.27	7310.90 ± 1451.06	0.132	0.985	0.01
AMD	8.18 ± 1.78	8.09 ± 1.62	7.50 ± 1.05	8.10 ± 1.61	0.360	0.875	0.02
Psychological variables							
Stress control	49.71 ± 10.72	49.15 ± 10.83	46.67 ± 21.50	47.57 ± 10.55	1.334	0.255	0.06
Influence of performance evaluation	30.14 ± 7.66	32.74 ± 8.26	24.17 ± 13.95	27.81 ± 7.22	2.598	0.069	0.11
Motivation	17.61 ± 4.38	15.91 ± 6.49	16.67 ± 4.37	18.86 ± 4.91	1.149	0.339	0.05
Mental ability	20.69 ± 4.76 *	16.82 ± 5.76 *	19.50 ± 5.09	20.00 ± 4.79	2.798	0.020	0.11
Team cohesion	15.59 ± 5.12	12.65 ± 7.09	14.33 ± 5.00	16.86 ± 4.76	2.180	0.062	0.09

* Differences are significant at the *p* < 0.05 level.

## Data Availability

Research data will be available upon request to the correspondence researcher.

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
