# Peer review of "Analysis of Differences in Injuries in Padel Players According to Sport-Specific Factors, Level of Physical Activity, Adherence to the Mediterranean Diet, and Psychological Status"

_sports, 2025, doi:10.3390/sports13070228_

Round 1

Reviewer 1 Report

Comments and Suggestions for Authors

Dear Authors,
The work sent to me for review is an important research topic. Its value also comes from the fact that it addresses aspects of the new sports discipline that is Padel.
The work requires several corrections and additions. Detailed suggestions for individual sections:

introduction:
you can try to include some photos showing the discipline
-you can also add a graph showing the heart rate during a match (especially since you describe it)
-it is also worth emphasizing the role of warm-up or the level of flexibility or general fitness as factors that can reduce the risk of injuries

material and methods:
in this section, please add a detailed graph illustrating the course of the research experiment, taking into account individual activities, number of participants and inclusion and exclusion criteria

results:
please highlight statistically significant results in tables
-it is worth presenting key results in a graphic form, which is always more interesting for potential recipients

the work after corrections is worth publishing
with respect

Author Response

Reviewer 1

Dear Authors,

- The work sent to me for review is an important research topic. Its value also comes from the fact that it addresses aspects of the new sports discipline that is Padel. The work requires several corrections and additions.

+ Dear reviewer, thank you very much for accepting to review our manuscript and for providing input to improve it. We will attend to all your requests to increase the scientific quality of it.

Detailed suggestions for individual sections:

Introduction:

- You can try to include some photos showing the discipline.

+ Thank you very much for your contribution. Photos of the court and its dimensions have been included to provide a better understanding of the sport.

- You can also add a graph showing the heart rate during a match (especially since you describe it)

+ Thank you very much for your contribution. This graph, along with complementary information to aid its understanding, has been included in the introduction

- It is also worth emphasizing the role of warm-up or the level of flexibility or general fitness as factors that can reduce the risk of injuries.

+ Thank you very much for this valuable contribution. This information has been included in the introduction to highlight the importance of injury prevention factors in this sport.

Material and methods:

- In this section, please add a detailed graph illustrating the course of the research experiment, taking into account individual activities, number of participants and inclusion and exclusion criteria.

+ Thank you very much for your recommendation. A flowchart containing all relevant information about the participants has been included.

Results:

- Please highlight statistically significant results in tables.

+ Thank you very much. Statistically significant values have been highlighted in bold in each table.

- It is worth presenting key results in a graphic form, which is always more interesting for potential recipients.

+ Thank you very much. You are absolutely right. We decided to present the chi-square tables in graphical form, as they are much more visually accessible.

- The work after corrections is worth publishing. With respect

+ Thank you once again for all the suggestions provided. We hope we have addressed all the requested changes and that the quality of the manuscript has improved.

Reviewer 2 Report

Comments and Suggestions for Authors

Based on a comprehensive evaluation across all criteria, this article demonstrates good overall merit with strong potential for impact in sports science and injury prevention, warranting publication with major revisions.

Presentation Barriers

 Table overload: 10+ tables with granular subgroups (e.g., 7 court types).

No visualizations: Fails to highlight key findings (e.g., footwear-risk link).

 Poor statistical communication: Uninterpreted effect sizes (e.g., Cohen’s *d*=6.18).

Methodological Gaps

Cross-sectional design: Prevents causal inference.

 Convenience sampling: Spanish bias (Murcia/Valencia).

 Low-reliability tools: MEDAS (α=0.51), CPRD subscales (α=0.34).

Revision Priorities

Visual Revolution

Add 3–4 figures:

Bar chart: Injury severity by footwear type.

Heatmap: Court surface × anatomical injury risk.

Infographic: Key prevention takeaways.

Statistical Clarity

Simplify tables (collapse court types → sand vs. non-sand).

Interpret effect sizes (e.g., "Cohen’s *d*=6.18 = very large effect").

Flag underpowered analyses (e.g., bone injuries, *n*=6).

Methodological Transparency

Explicitly state design limitations in Abstract/Conclusion.

Recommend prospective validation in future work.

Author Response

Reviewer 2

Presentation Barriers

- Table overload: 10+ tables with granular subgroups (e.g., 7 court types).

+ Thank you. We have reduced the number of tables and converted them into graphs.

- No visualizations: Fails to highlight key findings (e.g., footwear-risk link).

+ Thank you for your valuable suggestion. We have included bar charts to better illustrate key findings. These visualizations have been added to the Results section.

- Poor statistical communication: Uninterpreted effect sizes (e.g., Cohen’s *d*=6.18).

+ Thank you very much. The information regarding effect size has been included in both the Results and Discussion sections.

Methodological Gaps

- Cross-sectional design: Prevents causal inference.

+ We acknowledge that the cross-sectional design prevents causal inference. This limitation has been highlighted in the discussion, and we have suggested that future longitudinal research is needed to confirm potential causal relationships..

- Convenience sampling: Spanish bias (Murcia/Valencia).

+ Thank you very much for your suggestion. The Spanish bias has been included in the limitations section.

Low-reliability tools: MEDAS (α=0.51), CPRD subscales (α=0.34).

+ Thank you for your contribution. Although these instruments are generally considered valid and reliable, some dimensions show low reliability. This has been acknowledged in the limitations section, and we have suggested complementing these measures with others that provide additional information.

Revision Priorities

Visual Revolution

Add 3–4 figures:

Bar chart: Injury severity by footwear type.

Heatmap: Court surface × anatomical injury risk.

Infographic: Key prevention takeaways.

+ Thank you very much for your contribution. Some of the tables have been converted into text, and a figure displaying the significant results has been included in their place. We used bar charts in all cases, as the heatmap became too limited after collapsing the court surface categories. The infographic has been included as supplementary material.

Statistical Clarity

- Simplify tables (collapse court types → sand vs. non-sand).

+ Thank you very much for this valuable contribution. We have simplified this part of the tables, as it did not provide much information due to the small sample size for each court type. With this modification, the data are easier to interpret.

- Interpret effect sizes (e.g., "Cohen’s *d*=6.18 = very large effect").

+ Thank you very much. This information has been included in both the Results and Discussion sections.

- Flag underpowered analyses (e.g., bone injuries, *n*=6).

+ Thank you for your clarification. You are absolutely right about the importance of highlighting this information, as it is not generalizable. It has been noted in the Limitations sections.

Methodological Transparency

- Explicitly state design limitations in Abstract/Conclusion.

+ Thank you for your suggestion. This information has been included in both the abstract and the conclusion sections.

- Recommend prospective validation in future work.

+ Thank you very much. This point has been included in the Conclusion section.

+ Thank you again for your review. We hope that we have responded to your requests and have improved visually and the content of the manuscript.